# Blocking of SGLT2 to Eliminate NADPH-Induced Oxidative Stress in Lenses of Animals with Fructose-Induced Diabetes Mellitus

**DOI:** 10.3390/ijms23137142

**Published:** 2022-06-27

**Authors:** Ying-Ying Chen, Tsung-Tien Wu, Chiu-Yi Ho, Tung-Chen Yeh, Gwo-Ching Sun, Ching-Jiunn Tseng, Pei-Wen Cheng

**Affiliations:** 1Department of Ophthalmology, Kaohsiung Veterans General Hospital, Kaohsiung 81362, Taiwan; yychen@vghks.gov.tw (Y.-Y.C.); ttwu@vghks.gov.tw (T.-T.W.); 2School of Medicine, National Yang-Ming University, Taipei 11221, Taiwan; 3Department of Medical Education and Research, Kaohsiung Veterans General Hospital, Kaohsiung 81362, Taiwan; hochiuyi1987@gmail.com (C.-Y.H.); tseng@vghks.gov.tw (C.-J.T.); 4Department of Biomedical Sciences, National Sun Yat-sen University, Kaohsiung 80424, Taiwan; 5Department of Internal Medicine, Division of Cardiology, Kaohsiung Veterans General Hospital, Kaohsiung 81362, Taiwan; tcyeh9@gmail.com; 6Department of Anesthesiology, Kaohsiung Veterans General Hospital, Kaohsiung 81362, Taiwan; gcsun39@yahoo.com.tw; 7Department of Medical Research, China Medical University Hospital, China Medical University, Taichung 40402, Taiwan

**Keywords:** cataract, type 2 diabetes mellitus, NADPH oxidase, glucose transporter, resveratrol

## Abstract

Chronic hyperglycemia triggers an abnormal rise in reactive oxygen species (ROS) that leads to blindness in patients with diabetes mellitus (DM) and cataracts. In this study, the effects of dapagliflozin, metformin and resveratrol on ROS production were investigated in lens epithelial cells (LECs) of animals with fructose-induced DM. LECs were isolated from patients without DM, or with DM devoid of diabetic retinopathy. Animals were treated with 10% fructose for 8 weeks to induce DM, which was verified by monitoring blood pressure and serum parameters. For drug treatments, 1.2 mg/day of dapagliflozin was given for 2 weeks, 500 mg/kg/day of metformin was given, and 10 mg/kg/day of resveratrol was given. Dihydroethidium was used to stain endogenous O_2_˙^−^ production in vivo of the LECs. Superoxide production was expressed in the cataract of DM, or patients without DM. Sodium–glucose cotransporter 2 (SGLT2), glucose transporter 1 (GLUT1), GLUT5, the reduced form of nicotinamide adenine dinucleotide phosphate (NADPH) oxidase subunit p47/p67-phox, NOX4 and RAGE were significantly increased in LECs with DM. In addition, the dapagliflozin treatment reduced GLUT5, p47/p67-phox, NADPH oxidase 4 (NOX4) and receptor for advanced glycation end products (RAGE) expressions. On the contrary, metformin or resveratrol inhibited p47-phox, GLUT5, and SGLT2 expressions, but not nuclear factor erythroid 2–related factor 2 (NRF2). In summary, dapagliflozin, metformin or resveratrol down-regulated p47-phox expression through SGLT2 inactivation and ROS reduction. These important findings imply that SGLT2 can be blocked to ameliorate oxidative stress in the cataracts of DM patients.

## 1. Introduction

Diabetic patients aged above 65 may suffer irreversible cataract development; however, good control over the metabolism may reverse cataract development in younger diabetic patients [1]. The Wisconsin Epidemiologic Study of Diabetic Retinopathy reported that 24.9% of type 2 diabetes patients and 8.3% of type 1 diabetes patients had a history of 10-year-long cumulative incidence for cataract surgery [2]. Type 1 diabetics’ risk factors include age, severity of diabetic retinopathy (DR) and proteinuria; Type 2 diabetics’ risk factors include prolonged duration of diabetes, lack of metabolic control and use of insulin [1,3]. At present, cataract removal and intraocular lens implants are the major treatments for diabetic cataracts. Nevertheless, surgery may result to severe postoperative complications such as corneal edema, infection, and ocular hypertension, especially in the elderly and individuals having hyperglycemic conditions [4]. In consequence, alternative treatment for diabetic cataracts is imminent.

In the past 50 years, fructose consumption has surpassed 10% of our daily calorie intake in the form of sucrose and inexpensive corn-based sweeteners such as high-fructose corn syrup (HFCS) [5]. Even though fructose is not a healthy constituent for consumption, fructose is commonly found in the modern diet and has been the major culprit for metabolic diseases. In prior studies, modulation of fructose transporter GLUT5 (SLC2A5) was found to have potential for treating metabolic disease, since GLUT5 is involved in fructose metabolism and intestinal fructose absorption [6]. Most importantly, GLUT5 is the major fructose transporter in human eyes [7], whereas Mantych et al. suggested that GLUT1 is the main glucose transporter typically found in lens epithelial cells (LECs) of the blood-aqueous barrier. The LECs play an important role in preventing oxidative stress and nutrient transport across the aqueous humor. Surprisingly, GLUT1, GLUT5 and SGLT2 were previously found to be highly expressed in cataracts and LECs of DM rats [8,9].

Fructose promotes reactive oxygen species (ROS) production and downregulates key antioxidant enzymes such as superoxide dismutase (SOD) [10,11]. ROS overproduction was found to worsen diabetic complications including cataract development in patients having chronic hyperglycemia [12]. Recent studies showed that phagocyte-type NADPH oxidase, composed of two catalytic (p22-phox and p91-phox) and four regulatory subunits (p40-phox, p47-phox, p67-phox and Ras-related C3 botulinum toxin substrate 1 or Rac1), is the major factor leading to ROS production in the vasculature network [13]. Furthermore, NADPH oxidase is derived from advanced glycation end products (AGEs) during fructose metabolism, whose receptors is receptor for AGEs (RAGE). RAGE and AGE belong to senescent protein derivatives and are both associated with metabolic syndromes [14,15]. Signaling through RAGE and AGE may directly induce ROS via NADPH oxidases or through other unidentified mechanisms [16].

In this study, metformin was investigated because it is a biguanide commonly used to lower serum glucose in non-insulin-dependent diabetic patients, and it enhances glucose metabolism in the retina, protects retinal photoreceptors and retinal pigment epithelium from heritable mutations, and has been found to lower oxidative stress in a preclinical animal study [17]. Apart from metformin, resveratrol was also examined since it is an antioxidant, promotes 5′ adenosine monophosphate-activated protein kinase (AMPK) activity and improves insulin sensitivity to relieve various metabolic disorders [18,19].

Dapagliflozin, a sodium glucose cotransporter 2 (SGLT2) inhibitor, is a newly-emerging compound used to treat diabetes. A SGLT2 inhibitor acts on the proximal tubules to enhance sugar release into the urine, independent of insulin. SGLT2 inhibitors are known to be less likely to cause side effect such as hypoglycemia, compared to conventional treatment using insulin. We previously showed that GLUTs may be involved in RAGE-induced superoxide production and cataract formation in DM patients, as well as a type 2 DM animal model. Dapagliflozin may have been involved in inhibition of SGLT2 and GLUT expressions, downregulation of RAGE and NADPH oxidases, and suppression of ROS accumulation, thereby protecting the LECs [20]. Until now, neither the role of SGLT2 nor NADPH-dependent ROS in fructose-associated diabetic cataract development have been clarified. We hypothesized that ROS formation during fructose-induced diabetic cataract development requires SGLT2 and NADPH oxidase. By targeting NADPH oxidase (p47-phox) using dapagliflozin, metformin or resveratrol, SGLT2 was inactivated and ROS was reduced. Our findings support the theory that a SGLT2 blocker inhibits oxidative stress in the LECs, a powerful tool for studying cataract pathogenesis in DM patients.

## 2. Results

### 2.1. Superoxide Production in Cataract Lenses of DM Patients

Oxidized lenses and decreased repair ability are linked to increased lens opacity and cataract development during aging and in patients with chronic hyperglycemia [21,22]. Ten DM (−) patients with cataracts (4 males, 6 females), and DM (+) patients with cataracts (4 males, 6 females) were enrolled in this study with informed consent as control and experimental groups, respectively. The DM (−) patients’ mean age was 64.0 ± 4.9 years old, and DM (+) patients’ mean age was 67.0 ± 3.5 years old (*p* = 0.13). The HbA1c level was 7.70 ± 0.54 for the DM (+) patients (Table 1). DHE fluorescence assay was used to analyze superoxide in the LECs with and without DM (Figure 1A). Results showed that DHE levels were apparent in the LEC of diabetic and non-diabetic cataract sections (Figure 1A).

### 2.2. Several Drugs Prevent Fructose-Mediated Metabolic Defects

Table 2 provides detailed measurements for fasting glucose, triglyceride, high-density lipoprotein, and cholesterol levels in the animals. Consistent with a recent report, we observed significant elevation of serum triglyceride after fructose administration, compared to controls [11,23]. The fasting blood fructose level was also higher in fructose-fed animals compared to those fed no fructose, whereas the direct high-density lipoprotein level was significantly lower. Dapagliflozin, metformin and resveratrol administration for two weeks prevented fructose-mediated metabolic defects. The fasting glucose and triglyceride level were significantly lower, but direct high-density lipoprotein levels showed no difference in the dapagliflozin and resveratrol groups compared to the fructose only group. These results indicate that dapagliflozin, metformin and resveratrol suppressed fructose-induced DM. 

### 2.3. Dapagliflozin Inhibits RAGE-Induced NADPH Oxidase Subunit Production by Abolishing SGLT2 Expression in LECs of Fructose-Induced DM Rats 

Previously, we demonstrated that dapagliflozin downregulated RAGE-induced NADPH oxidase expression in LECs via inactivation of GLUTs and reduction in ROS generation. Here, we found that NADPH oxidase p47-phox, GLUT5, NOX4, and RAGE proteins levels were significantly increased compared to the control. Additionally, co-administration of dapagliflozin prevented this increase (Figure 2A,B). Immunoblotting analysis demonstrated that SGLT2, GLUT1 and RAGE protein expressions were decreased after dapagliflozin treatment in the diabetic animals (Figure 2C). Dapagliflozin also reduced NADPH oxidase p67-phox and NOX4 protein levels in the diabetic LECs (Figure 2D). Based on these findings, we suggest that dapagliflozin downregulated fructose-induced ROS through inhibiting SGLT2 signaling. 

### 2.4. Metformin Reduces NADPH Oxidase Subunit Production by Abolishing SGLT2 Expression in LECs of Fructose-Induced DM Rats 

Previously, we showed that dapagliflozin reduced ROS and downregulated RAGE-induced NADPH oxidase expression through GLUTs inactivation in the LECs. In this study, p47-phox and GLUT5 protein levels were significantly decreased after metformin addition (Figure 3A,B). Furthermore, metformin decreased SGLT2 and NADPH oxidase p67-phox protein expressions, and NRF2 was not affected (Figure 3C,D). As a result, metformin blocks SGLT2 signaling and downregulates ROS generation in the DM animals.

### 2.5. Resveratrol Improves SGLT2-Induced NADPH Oxidase Subunit Generation in LECs of Fructose-Induced DM Rats 

Our previous study demonstrated that dapagliflozin decreased ROS generation and downregulated RAGE-induced NADPH oxidase expression via GLUTs inactivation. Here, we found that resveratrol significantly lowers p47-phox and GLUT5 protein expressions in the LECs of DM animals (Figure 4A,B). Resveratrol decreased SGLT2 protein expression, but has no effect on NRF2 (Figure 4C,D). In consequence, resveratrol downregulates ROS generation by inhibiting SGLT2 function in the lens of diabetic animals.

## 3. Materials and Methods

### 3.1. Chemicals and Reagents

Drugs and chemicals were obtained from Sigma-Aldrich (Sigma Chemical Co., St. Louis, MO, USA). Primary antibodies against GLUT1, SGLT2 and RAGE were purchased from Abcam (Abcam, Cambridge, MA; ab115730, ab3611, ab65965, and ab37296, respectively), NADPH oxidase 4 (NOX 4) from Novus Biologicals (NB110-58849, Englewood, CO, USA), p67-phox from Millipore (07-502, Billerica, MA, USA), and GLUT5 from GeneTex (Long Beach, CA, USA). 

### 3.2. Ethics Statement

This study was reviewed and approved by the Institutional Review Board of Kaohsiung Veterans General Hospital (Kaohsiung, Taiwan; IRB number: VGHKS18-CT4-22).

This prospective study is comprised of patients who underwent phacoemulsification and intraocular lens implantation between August 2018 and September 2019 at Kaohsiung Veterans General Hospital. Protocols follow the guidelines from Declaration of Helsinki, approved by the hospital’s institutional review board. After giving a detailed explanation of the surgical procedures and possible complications, all participants provided written informed consent. All the data and specimens were collected and anonymized before analysis. The patients were selected based on clinically confirmed nuclear grade 3 cataracts according to the Lens Opacities Classification System III [17]. The subjects were classified into the following 2 groups: (1) patients without DM (Group 1); (2) patients with DM (Group 2).

### 3.3. Animals

Sixteen-week-old male WKY rats were obtained from the National Science Council Animal Facility (NSCFA) (Taipei, Taiwan) and housed in the animal facility at Kaohsiung Veterans General Hospital (VGHKS; Kaohsiung, Taiwan). NSCAF and VGHKS have received international certification from the AAALACi. Animals entering in the VGHKS were free of infectious organisms that are pathogenic and/or capable of interfering with research objectives. The rats were caged individually in a light- (12-h light/12-h dark cycle) and temperature-controlled (23–24 °C) room and fed with normal rat chow (Purina; St. Louis, MO, USA) and tap water ad libitum. All animal protocols are in accordance with the ARRIVE guidelines [24,25], approved by the Animal Research Committee and the Institutional Review Board at Kaohsiung Veterans General Hospital. Drug treatments for animals were conducted using a blinded method. 

The animals underwent 1 week of acclimatization before being subjected to blood pressure measurement for another 1 week. After a stabilization period, the rats were randomly assigned to 5 groups, comprising 6 rats per group with the following oral administration regime: (1) control: pure drinking water; (2) fructose: 10% fructose in drinking water for 8 weeks; (3) dapagliflozin: 10% fructose in drinking water for 6 weeks, followed by 2-week fructose + dapagliflozin (3 mg/kg/day); (4) metformin group: 10% fructose in drinking water for 6 weeks, followed by 2-week fructose + metformin (500 mg/kg/day); (5) resveratrol group: 10% fructose in drinking water for 6 weeks, followed by 2-week fructose + resveratrol (10 mg/kg/day). All rats in the experimental groups developed type 2 DM, and no incidence of heart failure or sudden death was found.

### 3.4. Tissue Collection

Based on the 2013 American Veterinary Medical Association (AVMA) guidelines, all animals were euthanized using 100% CO_2_. Within 2–5 min after the animals died, their lenses were removed and immediately frozen on dry ice. Tissues collected from the same experimental groups were pooled and stored at −80 °C.

### 3.5. Measurement of ROS in Lenses from DM Patients and LECs from Type 2 DM Rats

The endogenous in vivo O_2_^¯^ levels produced in humans with DM cataracts and fructose-fed rats were determined by staining the anterior region of the lens capsule with dihydroethidium (DHE; Invitrogen, Carlsbad, CA, USA). Lens epithelial cell (LEC)-containing slices removed from the rats were embedded in OCT (Shandon Cryomatrix; Thermo Electron Co., Pittsburgh, PA, USA), flash-frozen in a methylbutane ice-cold bath, and placed in liquid nitrogen. Lens capsular flaps were stained with 1 μM DHE for 20 min, away from light, at 37 °C in a 5% CO_2_ incubator. Samples were imaged using a fluorescence microscope and analyzed using the Zeiss LSM Image software (Carl Zeiss MicroImaging, Jena, Germany).

### 3.6. Measurement of Physiological Indices

In the final stage of the study, 1–2 mL of blood was collected from the experimental animals by cardiac puncture. Plasma glucose, direct high-density lipoprotein (dHDL), total cholesterol (TC) and triglyceride (TG) levels were determined using a clinical chemistry analyzer (Ortho Clinical VITROS™ 350 System, Rochester, NY, USA). 

### 3.7. Immunoblotting Analysis

Total protein extracts were prepared by homogenizing the lenses in a lysis buffer with protease and phosphatase inhibitor cocktails and then incubating for 1 h at 4 °C. The extracted proteins (assessed by BCA protein assay; Pierce) were subjected to 7.5–10% SDS-Tris glycine gel electrophoresis and transferred onto a polyvinylidene difluoride membrane (GE Healthcare, Buckinghamshire, UK).

The membrane was blocked in 5% non-fat skim milk using Tris-buffered saline/Tween-20 buffer (10 mmol/L Tris, 150 mmol/L NaCl, and 0.1% Tween-20, pH 7.4, slightly alkali), followed by incubation in anti-p67-phox (07-502), anti-GLUT1 (ab115730), rat anti-GLUT5 (GTX12098) and human anti-GLUT5 (GTX83626) primary antibodies at 1:1000 dilutions at 4 °C overnight. Peroxidase-conjugated anti-mouse or anti-rabbit secondary antibodies (1:5000 dilution) were used. Proteins were detected using an ECL-Plus detection kit from GE Healthcare and exposed to film. The developed films were scanned (photo scanner 4490, Epson, Long Beach, CA, USA) and analyzed using NIH image analysis software (National Institutes of Health, Bethesda, MD, USA).

### 3.8. Immunofluorescence Staining

Cryostat slices (10 μm) or the lens capsular flaps were incubated in anti-NOX4, anti-GLUT1, anti-GLUT5, anti-RAC1 or anti-RAGE primary antibodies (1:100 dilution). After a phosphate-buffered saline wash, sections were incubated in green-fluorescent Alexa Fluor 488- or 588-conjugated donkey anti-rabbit IgG (1:200 dilution; Invitrogen) at 25 °C for 2 h. The sections were analyzed using a fluorescence microscope and Zeiss LSM Image software (Carl Zeiss MicroImaging, DE Thüringen, Jena, Germany).

### 3.9. Statistical Analysis

All statistical analyses were carried out on raw data input using version 13.0, SPSS (SPSS Inc., Chicago, IL, USA). The DM and control groups were being compared using the non-parametric Mann-Whitney U test and the Chi-square test. One-way analysis of variance (ANOVA) with Scheffe’s post hoc comparison was performed to distinguish differences between groups. The *p* values < 0.05 were considered significant. Data represent mean ± standard error of the mean (SEM) of 6 independent experiments, each in triplicate. Data and statistical analysis comply with recommendations for experimental design and analysis in pharmacology [19,26].

## 4. Discussion

Metabolic disease is complicated, and metabolic-related complications have a crucial role in the growth rate of cataracts [27]. The cause of cataract development has been attributed to uncontrolled blood glucose level [28]. Lens opacity and oxidation [29], ROS formation, and sorbitol accumulation through AR conversion of glucose [30] have been linked to a distinctive rise in blood glucose and lens protein glycosylation during cataract development. Nevertheless, there has been no clear evidence to link anti-diabetic drugs and lowered cataract risks. Therefore, finding an effective method to treat diabetic cataract is necessary.

The sweetness of fructose makes it an ideal part of our modern diet despite its involvement in various metabolic diseases. Fructose requires fructose transporter GLUT5 for absorption in the intestine and GLUT5 is crucial for fructose metabolism, which makes GLUT5 a potential target for treating metabolic disease [6]. GLUT5 was reported to be the major transporter for fructose in human eyes [7]. Lim et al., observed that GLUT1 expression is prominent in the epithelial lining and in the fibrous region of both the rat and human lens, and SGLT2 is abundant in the core membrane, epithelium, outer and inner cortex in the rat lens [31]. Mantych et al., suggested that GLUT1 is the main transporter for glucose, which is typically located in the blood-aqueous barrier of the lens. Additionally, GLUT1, GLUT5 and SGLT2 are highly expressed in both DM cataracts and LECs of DM rats [8,9]. LECs play a crucial role in giving protection from oxidative stress and providing nutrient transport across the aqueous humor, and the energy acquired from glucose metabolism is required to maintain lens transparency. Within the lens tissue, glucose uptake is facilitated through members of the GLUT family, or sodium-dependent through the SGLT family, or both [32]. Chan et al., reported that SGLT2 presence in the bovine ciliary body epithelium may shed light on glucose transport, physiology of the bovine blood-aqueous barrier and glycemic control linked to diabetic cataract formation [33]. Previous studies demonstrated that SGLT2 and GLUT1 expressions in LECs were significantly elevated only in diabetic patients [20], indicating that SGLT2 and GLUT1 may be required for RAGE-induced superoxide generation and may be relevant to diabetic cataract formation. We found that SGLT2 and GLUT1 may be required for fructose-induced NADPH oxidase generation and pathogenesis of type 2 DM cataract development. The drugs dapagliflozin, metformin and resveratrol may have acted through suppression of SGLT2 and GLUT5 expressions, downregulated the RAGE and NADPH oxidase and prevented ROS accumulation, leading to protection from oxidative stress in the LECs (Figure 5).

Recently, SGLT2 inhibitors have been used to treat diabetes; however, their underlying mechanism is not yet understood. According to Zinman et al., empagliflozin, an SGLT2 inhibitor, reduces the risk of cardiovascular death, death from any cause, and hospitalization from heart failure in type 2 diabetic patients [34]. Cherney et al., reported that empagliflozin provides renal protection for type 1 diabetic patients [35]. SGLT2 inhibitors not only lower blood glucose but also suppress diabetic complications. Therefore, a SGLT2 inhibitor such as dapagliflozin is a novel therapeutic option for type 2 DM cataract treatment. The beneficial effects of dapagliflozin on the LECs may be mediated by downregulation of GLUT, RAGE and NADPH oxidases, and suppressed ROS accumulation (Figure 2). Additionally, metformin and resveratrol diminish ROS production through suppression of SGLT2 protein expression in the type 2 DM LECs (Figure 3 and Figure 4). We observed that dapagliflozin, metformin and resveratrol administration for 2 weeks prevented metabolic defects induced by fructose. Notably, the fasting glucose and triglyceride levels were significantly lower (Table 2). On the contrary, SGLT2 inhibitors did not lower insulin resistance or improve insulin secretion, which are the major pathological defects in type 2 DM [36]. Metformin is widely considered to be the optimal choice for type 2 DM treatment; however, long-term use of metformin seems to be less effective in overcoming diabetes-associated complications [37]. Resveratrol significantly improves insulin sensitivity and glucose homeostasis, therefore it is a novel addition to diabetes and its sequelae treatments [38]. Resveratrol, taken at a dose of 1–2 g/day, was found to improve glucose tolerance and post-meal plasma glucose in older adults having impaired glucose tolerance (IGT) [39]. Daily resveratrol oral supplementation for 3 months significantly reduces HbA1c, systolic blood pressure, total cholesterol, LDL-C, fasting blood glucose in type 2 DM patients [40]. Interestingly, Korshalm et al., demonstrated that individuals having modest insulin resistance show beneficial effects, but healthy individuals having normal glucose homeostasis will be less likely to be affected by resveratrol [41]. Further studies are necessary to determine the effect of resveratrol on chronic conditions such as insulin resistance.

Although dapagliflozin and metformin have been widely used to control DM-related complications, metabolic diseases are still escalating at an alarming rate, prompting further investigation into alternative therapies. However, a previous study showed that antidiabetic drug treatment did not reduce the risk of cataract. Food-derived bioactive compounds have been increasingly explored for their ameliorative effects against metabolic diseases. Our research team and others have extensively examined the beneficial effects of red wine, including its bioactive compounds such as resveratrol, in improving insulin sensitivity, and reducing oxidative stress, enhancing GLUT4 translocation, activating sirtuin 1 (SIRT1) and AMPK are all promising discoveries [19,42]. Furthermore, other studies found that SGLT2 inhibition by dapagliflozin concurrently enhanced renal gluconeogenesis. Theoretically, either increased insulin action or downregulated peroxisome proliferator-activated receptor gamma coactivator 1 alpha (PGC-1α) expression can inhibit forkhead box protein O1 (FoxO1), leading to a decrease in renal gluconeogenesis. A previous study indicated that resveratrol treatment upregulated components in renal insulin signaling at both gene and protein level in diabetic rats [43]. Finally, Sun et al., demonstrated that resveratrol significantly ameliorated dapagliflozin-induced renal gluconeogenesis by promoting the insulin signaling pathway which subsequently inhibits nuclear translocation of FoxO1 [44]. Hyperglycemia, a consequence of diabetes, enhances the formation of advanced glycation end products (AGEs) and senescent protein derivatives that result from the auto-oxidation of glucose and fructose [14]. AGE–RAGE interaction directly induces the generation of ROS via NADPH oxidases and/or other previously characterized mechanisms [16]. Furthermore, the dietary fructose-mediated generation of AGEs and activation of RAGE contribute to metabolic syndromes [15]. 

Notably, several clinical studies support the idea that resveratrol supplementation for patients already prescribed metformin could be equally effective in managing blood glucose and insulin, as well as systolic blood pressure [45]. Recent reports state that dapagliflozin, metformin or resveratrol downregulate NADPH oxidase subunit p47−phox expression via SGLT2 inactivation and ROS reduction. Based on this evidence, we suggest that resveratrol supplementation in patients taking either metformin or dapagliflozin could have an equally beneficial effect in managing diabetes-related complications, such as blood glucose and SGLT2 expression improvements, as well as cataract prevention. Nevertheless, these fascinating findings require further clinical study and validation.

## 5. Conclusions

In conclusion, we observed that DM rats’ LECs showed significantly increased expressions for SGLT2 or GLUT5 protein, and were inhibited by co-administration of dapagliflozin, metformin or resveratrol. This finding implies that resveratrol supplementation has immense promise for treating diabetic cataracts. However, the optimal delivery route for resveratrol supplementation requires further investigation.

## Figures and Tables

**Figure 1 ijms-23-07142-f001:**
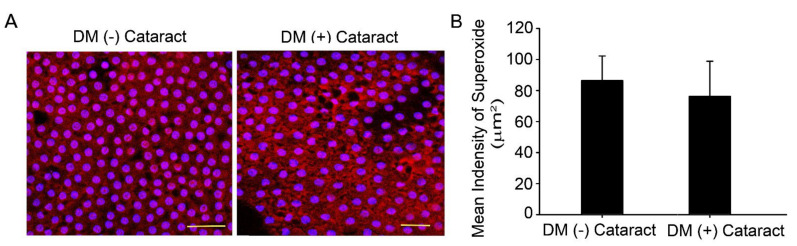
Superoxide generated in the lens of cataract in diabetic patients. (**A**) Representative images showing superoxide-positive cells in red, lens epithelial tissue was derived from cataracts of diabetic or non−diabetic patients. Cell nuclei were counterstained with DAPI (blue). Scale bar = 20 μm. (**B**) Quantified values (right) are represented as mean ± SEM (*n* = 10 for each group).

**Figure 2 ijms-23-07142-f002:**
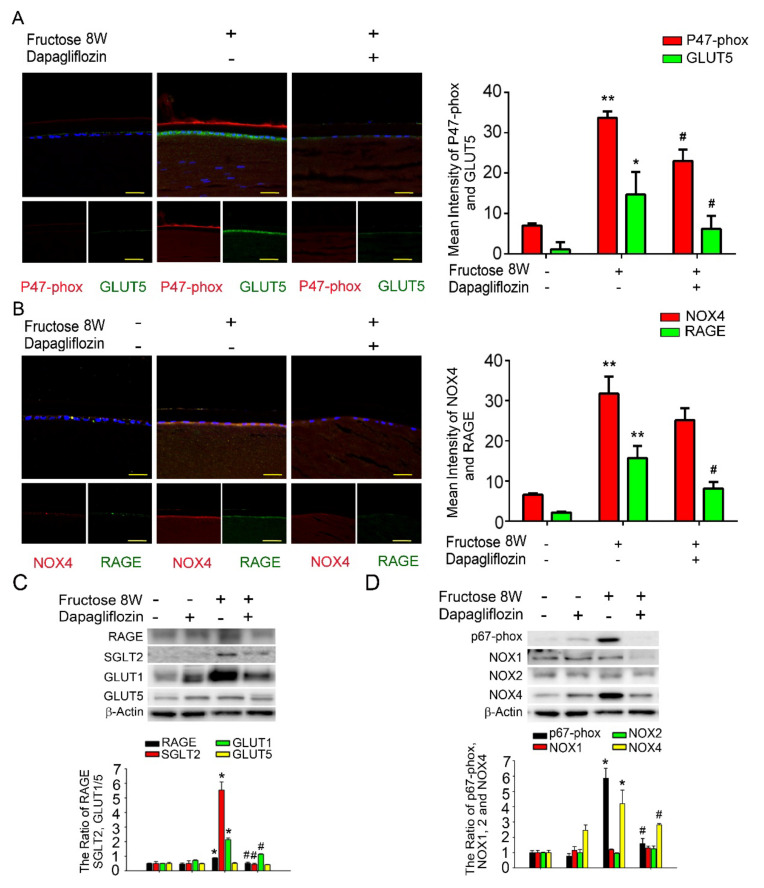
Fructose enhances RAGE expression through GLUT, and dapagliflozin reverses NADPH oxidase subunit (p67−phox) in the fructose-induced type 2 DM lens. (**A**,**B**) Representative images showing GLUT5− and RAGE-positive (green) and p47−phox and NOX4−positive (red) cells in the lens with or without dapagliflozin. Quantitative data were compared to no fructose group. Cell nuclei were counterstained in DAPI (blue). (**C**,**D**) RAGE, SGLT2, GLUT1, GLUT5, p67-phox and NOX1−4 protein expressions in the fructose-induced type 2 DM lens were significantly decreased by dapagliflozin. Values are presented as mean ± SEM (*n* = 6 for each group). Scale bar = 20 μm. * *p* < 0.05 and ** *p* < 0.001; ^#^
*p* < 0.05 versus Fructose 8 W.

**Figure 3 ijms-23-07142-f003:**
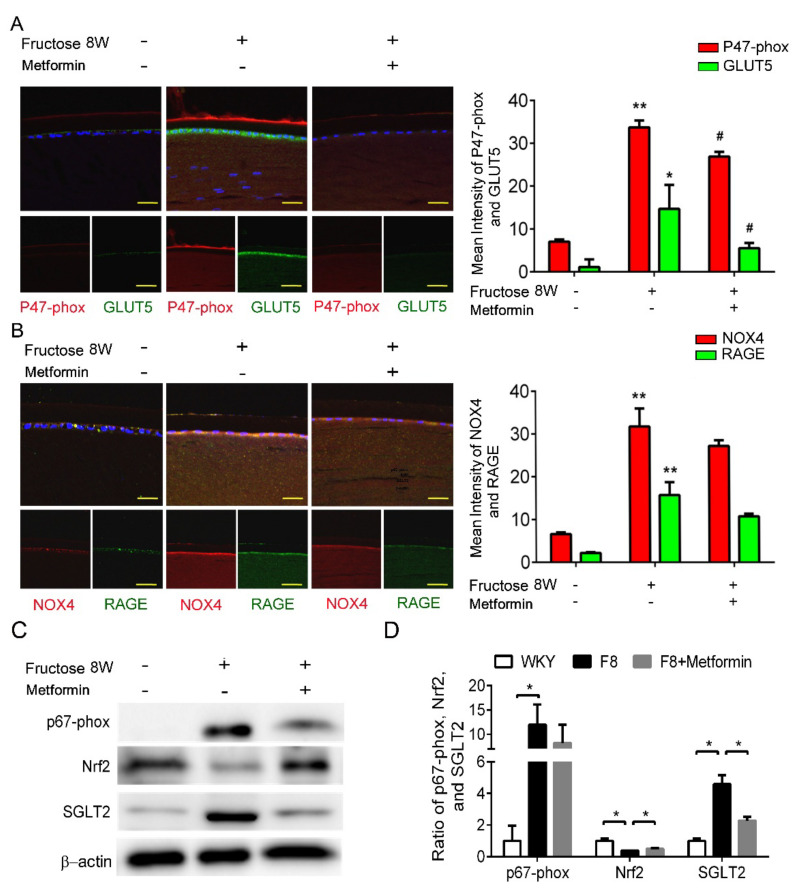
Metformin effectively reverses SGLT2-induced GLUT5 and NADPH oxidase subunit (p47−phox) in the fructose−induced type 2 DM lens. (**A**,**B**) Representative images showing GLUT5-positive (green) and p47−phox-positive (red) cells in the lens with or without metformin, compared to no fructose group. Cell nuclei are counterstained with DAPI (blue). (**C**,**D**) Quantitative analyses for p67−phox and SGLT2 after metformin was provided to the animals. Values are presented as mean ± SEM (*n* = 6 per group). Scale bar = 20 μm. * *p* < 0.05 and ** *p* < 0.001; ^#^
*p* < 0.05 versus Fructose 8 W.

**Figure 4 ijms-23-07142-f004:**
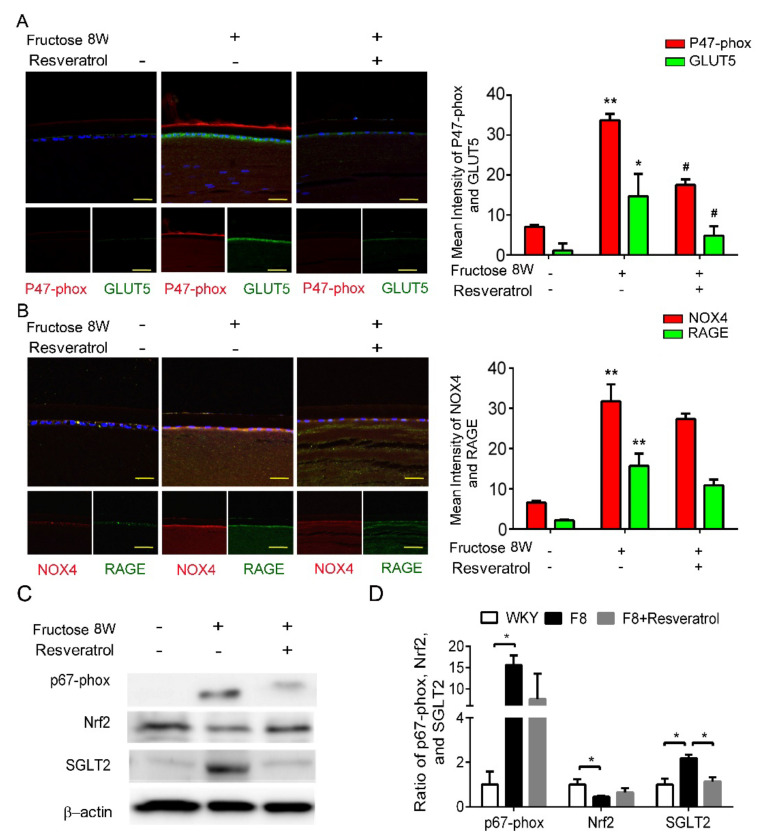
Resveratrol reduces fructose-induced NADPH oxidase subunit production mediated through a SGLT2−dependent mechanism in the epithelial section of type 2 DM lens. (**A**,**B**) Representative images for GLUT5−positive (green) and p47−phox NOX4-positive (red) cells in the lens after resveratrol treatment, compared to no fructose group. Cell nuclei were counterstained in DAPI (blue). (**C**,**D**) Quantitative analysis demonstrated that SGLT2 and p67−phox proteins` level were significantly decreased by resveratrol treatment. Values are presented as mean ± SEM (*n* = 6 per group). Scale bar = 20 μm. * *p* < 0.05 and ** *p* < 0.001; ^#^
*p* < 0.05 versus Fructose 8 W.

**Figure 5 ijms-23-07142-f005:**
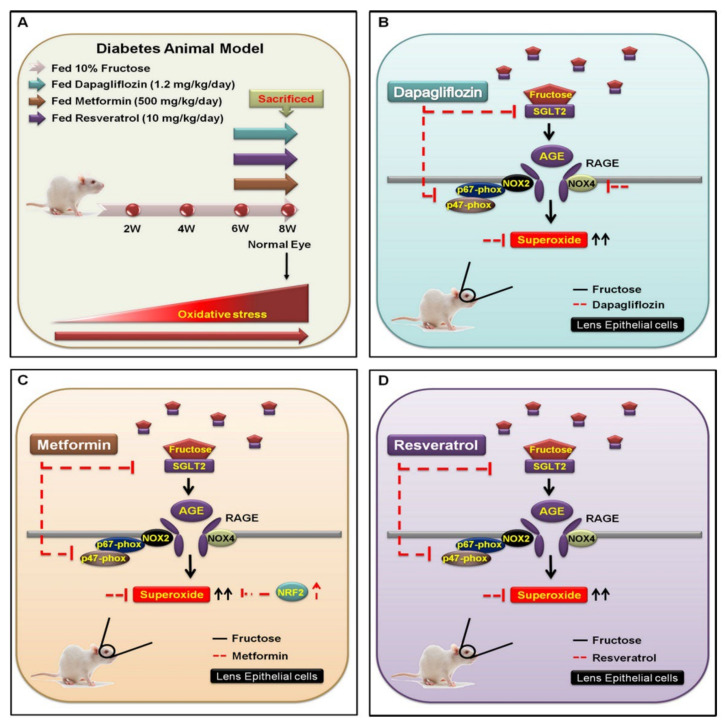
Underlying mechanism for inhibiting SGLT2−enchanced NOX2/4-dependent oxidative stress in the lens of type 2 DM. (**A**) WKY rats were categorized into four experimental groups: (1) 10% fructose only for 6 weeks; (2) followed by oral administration of dapagliflozin (1.2 mg/kg/day); (3) metformin (500 mg/kg/day); and (4) resveratrol (10 mg/kg/day) for 8 weeks. (**B**) The SGLT2 inhibitor dapagliflozin is an important fructose regulator that downregulates the SGLT2−induced activity of AGE−RAGE−NOX2/4 and p47−phox (red line). (**C**,**D**) Fructose requires SGLT2 to increase superoxide generation (black line); however, both metformin and resveratrol prevent superoxide accumulation through reduction of fructose-activated SGLT2 signal transduction (red line).

**Table 1 ijms-23-07142-t001:** Demographics and baseline clinical characteristics of the participants.

	Control Group (*n* = 10)	DM Group (*n* = 10)	*p* Value
Age (years)	64.0 ± 4.9	67.0 ± 3.5	0.13
Sex (Male:Female)	4:6	4:6	1.00
BMI	23.7 ± 2.8	25.4 ± 3.9	0.27
HbA1c	-	7.70 ± 0.54	

Hemoglobin A1c (HbA1c) level was determined in patients without DM (Control) and in DM patients without diabetic retinopathy (DM group). BMI: Body mass index. Values are shown as mean ± SEM.

**Table 2 ijms-23-07142-t002:** General characteristics of the experimental groups.

Parameter/Group	WKY	Fructose 8 W	Fructose 8 W+ Dapagliflozin	Fructose 8 W+ Metformin	Fructose 8 W+ Resveratrol
Fasting serum glucose (mg/dL)	94.3 ± 4.3	375.8 ± 10.8 *	298.3± 26.2 ^#^	324.8 ± 9.4 ^#^	311.7 ± 24.5 ^#^
dHDL (mg/dL)	67.7± 0.9	64.3 ± 0.3 *	61.6 ± 2.7	77.0 ± 1.0 ^#^	62.3 ± 1.5
Triglyceride (mg/dL)	103.5 ± 5.0	177.3 ± 5.2 *	152.3 ± 0.9 ^#^	124.0 ± 19.2 ^#^	149.0 ± 10.1 ^#^

The fasting serum glucose, high-density lipoprotein (HDL) and serum triglyceride (TG) levels were determined in 6 weeks of 10% fructose-fed animals, followed by fructose + dapagliflozin, metformin or resveratrol for 2 weeks. dHDL: direct high-density lipoprotein. The values are shown as mean ± SEM (*n* = 6 per group); * *p* < 0.05 versus control; ^#^
*p* < 0.05 versus Fructose 8 W.

## Data Availability

All data generated or analysed during this study are included in this published article.

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
