# Peer review of "Blocking of SGLT2 to Eliminate NADPH-Induced Oxidative Stress in Lenses of Animals with Fructose-Induced Diabetes Mellitus"

_ijms, 2022, doi:10.3390/ijms23137142_

Round 1

Reviewer 1 Report

Introduction

This section should be shortened somewhat. More emphasize the importance and rationale of the work.

Material and methods

Patients: give exact inclusion and exclusion criteria. How was the number of patients determined?

What diet did the animals receive? How was the concentration, form of administration, and timing of fructose determined? How was diabetes confirmed?

Describe in more detail how tissues were collected for the study.

Was ROS production measured immediately after tissue collection?

Why were NOX activity and protein expression at the mRNA level not assessed?

Results

Give more clinical data of the patients: blood biochemistry, glucose and insulin levels, lipid profile, etc.

Give basic data of the animals: body weight, BMI, food intake, insulin and HbA1c levels, HOMA-IR, etc.

LECs: provide details of the studies on this model. How were the cells obtained, how were they cultured, how were they processed, etc.?

A scale should be placed on each immunofluorescence staining image.

Show images of cataract lenses in animals.

Discussion

Overall, this is an observational study that does not provide mechanistic insight. I think the conclusion is poorly supported by the data, and the data are overinterpreted. Evaluation of insulin signaling and early/late protein glycation products is essential. 

Furthermore, the antiglycation/antioxidant effects of the drugs used should be emphasized more based on previous studies.

What are the limitations of the study?

Not all abbreviations are explained when first used.

Author Response

  1. This section should be shortened somewhat. More emphasize the importance and rationale of the work.

Reply:

Thanks for your comment. We agree with the reviewer`s suggestion that〝This section should be shortened somewhat, be more emphasized the importance and rationale of the work.〞. Therefore, we have reorganized the introduction structure and contents as the following: (P1-2).

  1. Introduction

Diabetic patients aged above 65 may suffer irreversible cataract development; however, good control over metabolism may reverse cataract development in younger diabetic patients (1). The Wisconsin Epidemiologic Study of Diabetic Retinopathy reported that 24.9% of type 2 diabetes, 8.3% of type 1 diabetes patients had a history of 10-year-long cumulative incidence for cataract surgery (2). Type 1 diabetic`s risk factors include age, severity of diabetic retinopathy (DR) and proteinuria; Type 2 diabetic`s risk factors include prolonged duration of diabetes, lack of metabolic control and use of insulin (1, 3). At present, cataract removal and intraocular lens implants are the major treatments for diabetic cataracts. Nevertheless, surgery may result to severe postoperative complications such as corneal edema, infection, and ocular hypertension, especially in the elderly and individuals having hyperglycemic conditions (4). In consequence, alternative treatment for diabetic cataracts is imminent.

In the past 50 years, fructose consumption has surpassed 10% of our daily calories intake in the form of sucrose and inexpensive corn-based sweeteners such as high-fructose corn syrup (HFCS) (5). Even though fructose is not a healthy constituent for consumption, but fructose is commonly found in modern diet and has been the major culprit for metabolic diseases. In prior studies, modulation of fructose transporter GLUT5 (SLC2A5) was found to have potential for treating metabolic disease since GLUT5 is involved in fructose metabolism and intestinal fructose absorption (6). Most importantly, GLUT5 is the major fructose transporter in human eyes (7), whereas Mantych et al. suggested that GLUT1 is the main glucose transporter typically found in lens epithelial cells (LECs) of the blood-aqueous barrier. The LECs plays important role in preventing oxidative stress and nutrient transport across the aqueous humor. Surprisingly, GLUT1, GLUT5 and SGLT2 were previously found to be highly expressed in cataracts and LECs of DM rats (8, 9).

Fructose promotes reactive oxygen species (ROS) production and downregulates key antioxidant enzymes such as superoxide dismutase (SOD) (10, 11). ROS overproduction was found to worsen diabetic complications including cataract development in patients having chronic hyperglycemia (12). Recent studies showed that phagocyte-type NADPH oxidase, composed of two catalytic (p22-phox and p91-phox), four regulatory subunits (p40-phox, p47-phox, p67-phox and Ras-related C3 botulinum toxin substrate 1 or Rac1), is the major factor leading to ROS production in the vasculature network (13). Furthermore, NADPH oxidase is derived from advanced glycation end products (AGEs) during fructose metabolism, whose receptor is receptor for AGEs (RAGE). RAGE and AGE belong to senescent protein derivatives, and are both associated with metabolic syndrome (14, 15). Signaling through RAGE and AGE may directly induce ROS via NADPH oxidases or through other unidentified mechanisms (16).

In this study, Metformin was investigated because it is a biguanide commonly used to lower serum glucose in non-insulin-dependent diabetic patients, and it enhances glucose metabolism in the retina, protects retinal photoreceptors and retinal pigment epithelium from heritable mutations, and it lowers oxidative stress in preclinical animal study (17). Apart from Metformin, Resveratrol was also being examined since it is an antioxidant, promotes 5′ adenosine monophosphate-activated protein kinase (AMPK) activity, improves insulin sensitivity to relieve various metabolic disorders (18, 19).

Dapagliflozin, sodium glucose cotransporter 2 (SGLT2) inhibitor, is a newly-emerging compound used to treat diabetes. SGLT2 inhibitor acts on the proximal tubules to enhance sugar release into the urine, independent of insulin. SGLT2 inhibitor is known to less likely cause side effect such as hypoglycemia compared to conventional treatment using insulin. We previously showed that GLUTs may be involved in RAGE-induced superoxide production and cataract formation in DM patients, as well as type 2 DM animal model. Dapagliflozin may have been involved in inhibition of SGLT2 and GLUT expressions, downregulation of RAGE and NADPH oxidases, suppression of ROS accumulation, thereby protecting the LECs (20). Until now, neither the role of SGLT2 nor NADPH-dependent ROS in fructose-associated diabetic cataract development have been clarified. We hypothesized that ROS formation during fructose-induced diabetic cataract development requires SGLT2 and NADPH oxidase. By targeting NADPH oxidase (p47-phox) using Dapagliflozin, Metformin or Resveratrol, SGLT2 was inactivated and ROS was reduced. Our findings support that SGLT2 blocker inhibits oxidative stress in the LECs, a powerful tool for studying cataract pathogenesis in DM patients.

Material and methods

  1. Patients: give exact inclusion and exclusion criteria. How was the number of patients determined?

Reply:

This prospective study comprised patients who underwent phacoemulsification and intraocular lens implantation between February 2016 and February 2017 at Kaohsiung Veterans General Hospital. Written informed consent was obtained from each patient and/or guardian. After receiving a full explanation of the surgical procedures and possible complications, all patients provided written informed consent. The patients were selected based on clinically observable nuclear cataracts of grade 2 or 3 according to the Lens Opacities Classification System III [32]. Exclusion criteria included cataract hardness greater than grade 3. Patients with type 1 DM, rheumatologic disease, or any other systemic disease except type 2 DM were also excluded. The patients were classified into two groups: patients without DM and patients with DM but without DR. Data on DM durations and glycated hemoglobin A1c (HbA1c) levels were also evaluated. Based on DM group must have inclusion condition with HbA1c data, therefore, we selected the sample size match sample between control and DM group.

  1. What diet did the animals receive? How was the concentration, form of administration, and timing of fructose determined? How was diabetes confirmed?

Reply:

Thank you for your comment. In previous study, our data revealed significant elevation of SBP, MBP, serum LDL, triglyceride concentrations in the fructose group (10% fructose in drinking water for 4 weeks) compared to control and showed higher fasting blood glucose and fructose levels in the fructose group. Furthermore, the serum insulin and homeostatic model assessment as an index of insulin resistance (HOMA-IR) showed elevated levels in the fructose-fed rats, whereas the dHDL showed significantly decreased levels, which indicated that fructose induced type 2 DM model (11, 21).

Control

Fructose 4 week

Body weight (gm)  

Systolic blood pressure (mmHg)

Mean blood pressure (mmHg)

Fasting serum glucose (mg/dL)

Fasting serum fructose (mg/dL)

Fasting serum insulin (pmol/L)

HOMA-IR Index

dHDL (mg/dL)

LDL (mg/dL)

Triglyceride (mg/dL)

Cholesterol (mg/dL)

233 ± 6.6

128.4 ± 2.6

113.7 ± 0.94

110 ± 2.35

3.7±0.29

9.38 ± 0.36

0.38 ± 0.02

68.8 ± 0.67

27.8 ± 1.01

56.4 ± 3.0

96.6 ± 1.78

227.5 ± 10.3

151.9 ± 2.3*

135.8 ± 1.7*

240.3 ± 12.7*

14.1±0.09*

175.7 ± 24.5*

13.1 ± 2.9*

55.0 ± 1.00*

45.7± 0.28*

111.5 ± 8.96*

100.63 ± 3.97

  1. Describe in more detail how tissues were collected for the study.

Reply:

Thanks for the reviewer`s question on how patient samples were collected. The central flap of anterior lens capsular consisted a single layer of lens epithelium with apices directed inward, and a basal laminar which forms the lens capsule isolating the lens constituents. When patients had undergone phacoemulsification, the central flap of anterior lens capsular was taken from the patients. In Figure 1, we took the anterior region of lens capsule (central epithelium region like the red area) from patient and performed IF staining on it.

  1. Was ROS production measured immediately after tissue collection?

Reply:

We thank the reviewer positive comments. Collecting clinical samples from the Biobank would pose difficulty in ROS measurement immediately after tissue collection. Dihydroethidium (DHE), have been used extensively in tissue culture experiments to evaluate reactive oxygen species (ROS) production (22). Besides, we also detected NADPH oxidase submit protein expressions by western blot analysis and immunofluorescence to confirm oxidative stress in the LECs.

  1. Why were NOX activity and protein expression at the mRNA level not assessed?

Results

Reply:

Thank you for your valuable comment. We agree with the reviewer`s point about NOX activity and protein expression at the mRNA level. However, the lens capsules from the patient or rats were not enough both (protein concentration small than 10 ng/ul) for western blot and real-time PCR at the same time. Therefore, we used immunofluorence staining to instead real-time PCR. Besides, we have research funding and time limitation of the studies, Therefore, in the future, we will aim to check the NOX activity and RNA levels, which will improve the quality of our work. 

  1. Give more clinical data of the patients: blood biochemistry, glucose and insulin levels, lipid profile, etc.

Reply:

Because all the data and clinical specimens from Biobank were anonymized, we did not have information regarding of blood biochemistry, glucose and insulin levels, lipid profile, etc.

  1. Give basic data of the animals: body weight, BMI, food intake, insulin and HbA1c levels, HOMA-IR, etc.

Reply:

Thanks for the valuable suggestions of the reviewer. At present, we have revealed a significant elevation of water intake, fasting serum glucose in the fructose group (10% fructose in drinking water for 8 weeks) compared to the control, whereas the food intake showed significantly decreased level. Furthermore, the serum insulin and homeostatic model assessment as an index of insulin resistance (HOMA-IR) showed elevated levels in the fructose-fed rats, indicating that fructose had induced type 2 DM model. Thus, a glycosylated hemoglobin lab value provides average blood glucose levels more than 3-month period. The HBA1c is an important part of long-term blood glucose monitoring. However, we applied a 8-week feeding regime in the animal study which is not long term. In consequence, the HbA1c levels did not show any difference (as shown in the Table 1).

Table 1.

Control (N=5)

Fructose 8 week (N=5)

Mean±SEM

Mean±SEM

Body weight (gm)

343.4 ± 9.4

334.7 ± 3.8

Food intake (gm/day)

16.7 ± 0.2

9.6 ± 0.2**

Water intake (gm/day)

31.5 ± 0.8

66.1 ± 2.7**

Fasting serum glucose (mg/dL)

78.3 ± 2.2

168.3 ± 4.7**

Insulin uIU/mL

4.4±0.4

15.76 ± 0.9**

HOMA-IR Index

1.7 ± 0.4

6.2 ± 0.3**

HbA1c (%)

4.5 ± 0.04

4.7 ± 0.10

** p<0.001 vs WKY group.

LECs: provide details of the studies on this model. How were the cells obtained, how were they cultured, how were they processed, etc.?

Reply:

Highly appreciate the reviewer`s comment. First question about how patient samples were collected? The central flap of anterior lens capsular consisted a single layer of lens epithelium with apices directed inward, and a basal laminar which forms the lens capsule isolating the lens constituents. When patients had undergone phacoemulsification, the central flap of anterior lens capsular was taken from the patients. In the figure 1, we took the anterior region of lens capsule (central epithelium region like the red area) from patient and performed IF staining to analyze it.

Figure 1.

  1. A scale should be placed on each immunofluorescence staining image.

Reply:

Thank you for your comments and suggestions. We have added the scale bar (20 mm) to Figures 1, 2, 3, and 4 of the revised manuscript.

  1. Show images of cataract lenses in animals.

Reply:

Thanks for the critical suggestions of the reviewer. Our previous finding showed lenses with maintained transparency; the squares under the lens were clearly visible in both the control and the fructose groups (20). To determine whether fructose-dependent ROS generation occurred in LECs of rats with fructose-induced type 2 DM, we examined the fructose effects on ROS levels. DHE fluorescence was used to measure superoxide levels in LECs from rats with fructose-induced type 2 DM after 8 weeks; representative images are shown in the next page figure 2A). DHE fluorescence was significantly increased in LECs of fructose-fed rats compared to the control group. However, dapagliflozin reversed these effects (figure 2B).

Figure 2.

  1. Furthermore, the antiglycation/antioxidant effects of the drugs used should be emphasized more based on previous studies.

Reply:

Thank you for your kind suggestion. We have reorganized the Discussion section as follows, and included the consequences for public health in the revised manuscript: (P9-12).

Discussion

Metabolic disease is complicated, and metabolic-related complications have crucial role in growth rate of cataract (23). The cause of cataract development has been attributed to uncontrolled blood glucose level (24). Lens` opacity and oxidation (25), ROS formation, and sorbitol accumulation through AR conversion of glucose (26) have been linked to distinctive rise in blood glucose and lens` protein glycosylation during cataract development. Nevertheless, there has been no clear evidence to link anti-diabetic drug and lowered cataract risks (27). Therefore, finding effective method to treat diabetic cataract is necessary.

The sweetness of fructose makes it an ideal part of our modern diet despite its involvement in various metabolic diseases. Fructose requires fructose transporter GLUT5 for absorption in the intestine and GLUT5 is crucial for fructose metabolism, which makes GLUT5 a potential target for treating metabolic disease (6). GLUT5 was reported to be the major transporter for fructose in human eyes (7). Lim et al. observed that GLUT1 expression is prominent in the epithelial lining and in the fibrous region of both rat and human lens, SGLT2 is abundant in the core membrane, epithelium, outer and inner cortex in the rat lens (28). Mantych et al. suggested that GLUT1 is the main transporter for glucose which is typically located in the blood-aqueous barrier of lens. Besides, GLUT1, GLUT5 and SGLT2 are highly expressed in both DM cataracts and LECs of DM rats (8, 9). LECs play crucial role in giving protection from oxidative stress and providing nutrient transport across the aqueous humor, and energy acquired from glucose metabolism is required to maintain lens transparency. Within the lens tissue, glucose uptake is facilitated through members of the GLUT family, or sodium-dependent through the SGLT family, or both (29). Chan et al. reported that SGLT2 presence in the bovine ciliary body epithelium may shed light on glucose transport, physiology of the bovine blood-aqueous barrier and glycemic control linked to diabetic cataract formation (30). Previous studies demonstrated that SGLT2 and GLUT1 expressions in LECs were significantly elevated only in diabetic patients (20), indicating that SGLT2 and GLUT1 may be required for RAGE-induced superoxide generation and may be relevant to diabetic cataract formation. We found that SGLT2 and GLUT1 may be required for fructose-induced NADPH oxidase generation and pathogenesis of type 2 DM cataract development. The drugs Dapagliflozin, Metformin and Resveratrol may have acted through suppression of SGLT2 and GLUT5 expressions, downregulated the RAGE and NADPH oxidase, prevented ROS accumulation, leading to protection from oxidative stress in the LECs (Fig.5).

Recently, SGLT2 inhibitors have been used to treat diabetes; however, its underlying mechanism is not yet understood. According to Zinman et al., Empagliflozin, an SGLT2 inhibitor, reduces the risk of cardiovascular death, death from any cause, and hospitalization from heart failure in type 2 diabetic patients (31). Cherney et al. reported that Empagliflozin provides renal protection for type 1 diabetic patients (32). SGLT2 inhibitors not only lower blood glucose but also suppress diabetic complications. Therefore, SGLT2 inhibitor such as Dapagliflozin is a novel therapeutic option for type 2 DM cataract treatment. The beneficial effects of Dapagliflozin on the LECs may be mediated by downregulation of GLUT, RAGE and NADPH oxidases, and suppressed ROS accumulation (Fig. 2). Besides, Metformin and Resveratrol diminish ROS production through suppression of SGLT2 protein expression in the type 2 DM LECs (Fig. 3-4). We observed that Dapagliflozin, Metformin and Resveratrol 2-week administration prevented metabolic defect induced by fructose. Notably, the fasting glucose and triglyceride levels were significantly lower (Table 2). On the contrary, SGLT2 inhibitors did not lower insulin resistance or improve insulin secretion, which are the major pathological defects in type 2 DM (33). Metformin is widely considered to be the optimal choice for type 2 DM treatment; however, long-term use of Metformin seems less effective to overcome diabetes-associated complications (34). Resveratrol significantly improves insulin sensitivity and glucose homeostasis, causing it a novel addition to diabetes and its sequelae treatments (35). Resveratrol, if taken 1-2 g/day was found to improve glucose tolerance and post-meal plasma glucose in older adults having impaired glucose tolerance (IGT) (36). Daily Resveratrol oral supplementation for 3 months significantly reduces HbA1c, systolic blood pressure, total cholesterol, LDL-C, fasting blood glucose in type 2 DM patients (37).  Interestingly, Korshalm et al., demonstrated that individuals having modest insulin resistance show beneficial effects, but healthy individuals having normal glucose homeostasis will be less likely affected by Resveratrol (38). Further studies are necessary to determine Resveratrol effect on chronic condition such as insulin resistance.

Albeit Dapagliflozin and Metformin have been widely used to control DM-related complications, metabolic diseases are still escalating at an alarming rate, prompting further investigation into alternative therapies. However, previous study showed that antidiabetic drug treatment did not show sign of lowering cataract development risk (27). Food-derived bioactive compounds have been increasingly explored for their ameliorative effects against metabolic diseases. Our research team and others have extensively examined the beneficial effects of red wine, including its bioactive compounds such as Resveratrol in improving insulin sensitivity, reducing oxidative stress, enhancing GLUT4 translocation, activating sirtuin 1 (SIRT1) and AMPK are all promising discoveries (19, 39). Furthermore, other studies found that SGLT2 inhibition by Dapagliflozin concurrently enhanced renal gluconeogenesis. Theoretically, either increased insulin action or downregulated peroxisome proliferator-activated receptor gamma coactivator 1 alpha (PGC-1α) expression can inhibit forkhead box protein O1 (FoxO1), leading to decrease in renal gluconeogenesis. Previous study indicated that Resveratrol treatment upregulated components in renal insulin signaling at both gene and protein level in diabetic rats (40). Finally, Sun et al. demonstrated that Resveratrol significantly ameliorated Dapagliflozin-induced renal gluconeogenesis by promoting the insulin signaling pathway which subsequently inhibits nuclear translocation of FoxO1 (41). Hyperglycemia, a consequence of diabetes, enhances the formation of advanced glycation end products (AGEs) and senescent protein derivatives that result from the auto-oxidation of glucose and fructose [14]. AGE–RAGE interaction directly induces the generation of ROS via NADPH oxidases and/or other previously characterized mechanisms (16). Furthermore, the dietary fructose-mediated generation of AGEs and activation of RAGE contribute to metabolic syndrome (15).

Notably, several clinical studies support that Resveratrol supplementation for patients already prescribed to Metformin could be equally effective in managing blood glucose, insulin, as well as systolic blood pressure (42). Recent reports state that Dapagliflozin, Metformin or Resveratrol downregulates NADPH oxidase subunit p47-phox expression via SGLT2 inactivation and ROS reduction. Based on this evidence, we suggest that Resveratrol supplementation in patients on either Metformin or Dapagliflozin could have equal beneficial effect in managing diabetes-related complications, such as blood glucose and SGLT2 expression improvements, as well as cataract prevention. Nevertheless, these fascinating findings require further clinical study and validation.

  1. What are the limitations of the study?

Reply:

We agree with the reviewer`s comment about the limitation of the studies, following are the limitations of this study:

Currently, surgery for cataract removal and intraocular lens implants are the main treatments for diabetic cataracts. However, surgery may result in numerous severe postoperative complications, including infection, corneal edema and increased intraocular pressure, especially in elderly people and those with hyperglycemic conditions [13]. Therefore, effective therapeutic strategies must be developed for the prevention and treatment of diabetic cataracts. However, investigators interested in performing small animal experiments involving DM related research, including pancreatic islet transplantation studies, have a variety of methods to choose from. These methods include but are not limited to models of DM that are spontaneous or genetically derived, chemically induced, diet induced, surgically induced, and now transgenic or knock-out derived. A single large dose of STZ is used for experiments attempting to cause severe T1DM by direct toxicity to β cells. Large doses can cause near total destruction of β cells and little or no measurable insulin production (Hayashi et al., 2006). However, at the end of 2010, the World Health Organization warned that high fructose consumption, mainly in the form of sweetened beverages, is a risk factor for several metabolic diseases (Aller et al., 2011). Fructose-fed rats are a model of acquired systolic hypertension that displays numerous features of metabolic syndrome in humans (Tran et al., 2009). High ROS levels directly disturb the physiological functions of cellular macromolecules and subsequently lead to lens pacification. Therefore, we used fructose-induced DM instead of ZDF rats or STZ-induced DM. There is accumulating evidence that fructose promotes an oxidative imbalance by simultaneously enhancing ROS production and down-regulating key antioxidant enzymes, such as superoxide dismutase (SOD)[14, 15]. Previous result showed photograph of a lens whose transparency is maintained; the squares under the lens are clearly visible in the normal and fructose groups. Dapagliflozin treatment reduced GLUT5, p47/p67-phox, NADPH oxidase 4 (NOX4) and RAGE expressions in the LECs of fructose induced-type 2 DM model. On the contrary, Metformin or Resveratrol inhibited p47-phox, GLUT5, and SGLT2 expressions, but not nuclear factor erythroid 2–related factor 2 (NRF2). Besides, we observed that DM rats` LECs showed significantly increased expressions for SGLT2 or GLUT5 protein, and were inhibited by co-administration of Dapagliflozin, Metformin or Resveratrol. This finding implies that Resveratrol supplementation has immense prospect for treating diabetic cataracts. However, the optimal delivery route for Resveratrol sup-plementation requires further investigation. In the future, we will check the effect of Dapagliflozin, Metformin or Resveratrol in STZ-induced DM, find out which will improve the quality of our work.  Our findings provide further insights into the pathogenesis of diabetic cataracts, inform clinical studies investigating the association between diabetes and cataract development, and support current treatments for cataracts in patients with diabetes. Besides, since all of the data and clinical specimens from Biobank were anonymized before we could access it for analysis, we did not have information regarding classify these participants were induced by fructose, these were the limitation of the studies.

  1. Not all abbreviations are explained when first used.

Reply:

Thank you for your kindly suggestion. We added the abbreviations when first used in the revised manuscript. (P1, Line 31-32,33-34; P1, Line 359, 362-363; P12, Line 393, 402).

References:

  1. Srinivasan S, Raman R, Swaminathan G, Ganesan S, Kulothungan V, Sharma T. Incidence, Progression, and Risk Factors for Cataract in Type 2 Diabetes. Invest Ophthalmol Vis Sci. 2017;58(13):5921-9.
  2. Kiziltoprak H, Tekin K, Inanc M, Goker YS. Cataract in diabetes mellitus. World J Diabetes. 2019;10(3):140-53.
  3. Peto T, Sandi F, Kumar V. Making the most of cataract surgery in patients with diabetes. Community Eye Health. 2019;31(104):80-1.
  4. Kumar PA, Reddy PY, Srinivas PN, Reddy GB. Delay of diabetic cataract in rats by the antiglycating potential of cumin through modulation of alpha-crystallin chaperone activity. J Nutr Biochem. 2009;20(7):553-62.
  5. DiNicolantonio JJ, O'Keefe JH, Lucan SC. Added fructose: a principal driver of type 2 diabetes mellitus and its consequences. Mayo Clin Proc. 2015;90(3):372-81.
  6. Zwarts I, van Zutphen T, Kruit JK, Liu W, Oosterveer MH, Verkade HJ, et al. Identification of the fructose transporter GLUT5 (SLC2A5) as a novel target of nuclear receptor LXR. Sci Rep. 2019;9(1):9299.
  7. Mantych GJ, Hageman GS, Devaskar SU. Characterization of glucose transporter isoforms in the adult and developing human eye. Endocrinology. 1993;133(2):600-7.
  8. Wu TT, Chen YY, Chang HY, Kung YH, Tseng CJ, Cheng PW. AKR1B1-Induced Epithelial-Mesenchymal Transition Mediated by RAGE-Oxidative Stress in Diabetic Cataract Lens. Antioxidants (Basel). 2020;9(4).
  9. Wu TT, Chen YY, Ho CY, Yeh TC, Sun GC, Tseng CJ, et al. 3H-1,2-Dithiole-3-Thione Protects Lens Epithelial Cells against Fructose-Induced Epithelial-Mesenchymal Transition via Activation of AMPK to Eliminate AKR1B1-Induced Oxidative Stress in Diabetes Mellitus. Antioxidants (Basel). 2021;10(7).
  10. Francini F, Castro MC, Schinella G, Garcia ME, Maiztegui B, Raschia MA, et al. Changes induced by a fructose-rich diet on hepatic metabolism and the antioxidant system. Life Sci. 2010;86(25-26):965-71.
  11. Yeh TC, Liu CP, Cheng WH, Chen BR, Lu PJ, Cheng PW, et al. Caffeine intake improves fructose-induced hypertension and insulin resistance by enhancing central insulin signaling. Hypertension. 2014;63(3):535-41.
  12. Zhang S, Chai FY, Yan H, Guo Y, Harding JJ. Effects of N-acetylcysteine and glutathione ethyl ester drops on streptozotocin-induced diabetic cataract in rats. Mol Vis. 2008;14:862-70.
  13. Bendall JK, Rinze R, Adlam D, Tatham AL, de Bono J, Wilson N, et al. Endothelial Nox2 overexpression potentiates vascular oxidative stress and hemodynamic response to angiotensin II: studies in endothelial-targeted Nox2 transgenic mice. Circulation research. 2007;100(7):1016-25.
  14. Guglielmotto M, Aragno M, Tamagno E, Vercellinatto I, Visentin S, Medana C, et al. AGEs/RAGE complex upregulates BACE1 via NF-kappaB pathway activation. Neurobiol Aging. 2012;33(1):196 e13-27.
  15. Miller A, Adeli K. Dietary fructose and the metabolic syndrome. Curr Opin Gastroenterol. 2008;24(2):204-9.
  16. Wautier MP, Chappey O, Corda S, Stern DM, Schmidt AM, Wautier JL. Activation of NADPH oxidase by AGE links oxidant stress to altered gene expression via RAGE. Am J Physiol Endocrinol Metab. 2001;280(5):E685-94.
  17. Xu L, Kong L, Wang J, Ash JD. Stimulation of AMPK prevents degeneration of photoreceptors and the retinal pigment epithelium. Proc Natl Acad Sci U S A. 2018;115(41):10475-80.
  18. Hawley SA, Gadalla AE, Olsen GS, Hardie DG. The antidiabetic drug metformin activates the AMP-activated protein kinase cascade via an adenine nucleotide-independent mechanism. Diabetes. 2002;51(8):2420-5.
  19. Bagul PK, Banerjee SK. Application of resveratrol in diabetes: rationale, strategies and challenges. Curr Mol Med. 2015;15(4):312-30.
  20. Chen YY, Wu TT, Ho CY, Yeh TC, Sun GC, Kung YH, et al. Dapagliflozin Prevents NOX- and SGLT2-Dependent Oxidative Stress in Lens Cells Exposed to Fructose-Induced Diabetes Mellitus. Int J Mol Sci. 2019;20(18).
  21. Cheng PW, Lin YT, Ho WY, Lu PJ, Chen HH, Lai CC, et al. Fructose induced neurogenic hypertension mediated by overactivation of p38 MAPK to impair insulin signaling transduction caused central insulin resistance. Free Radic Biol Med. 2017;112:298-307.
  22. Owusu-Ansah E, Banerjee U. Reactive oxygen species prime Drosophila haematopoietic progenitors for differentiation. Nature. 2009;461(7263):537-41.
  23. Pollreisz A, Schmidt-Erfurth U. Diabetic cataract-pathogenesis, epidemiology and treatment. J Ophthalmol. 2010;2010:608751.
  24. Tan AG, Kifley A, Holliday EG, Klein BEK, Iyengar SK, Lee KE, et al. Aldose Reductase Polymorphisms, Fasting Blood Glucose, and Age-Related Cortical Cataract. Invest Ophthalmol Vis Sci. 2018;59(11):4755-62.
  25. Stitt AW. Advanced glycation: an important pathological event in diabetic and age related ocular disease. Br J Ophthalmol. 2001;85(6):746-53.
  26. Ramana KV. ALDOSE REDUCTASE: New Insights for an Old Enzyme. Biomol Concepts. 2011;2(1-2):103-14.
  27. . !!! INVALID CITATION !!! (7).
  28. Lim JC, Perwick RD, Li B, Donaldson PJ. Comparison of the expression and spatial localization of glucose transporters in the rat, bovine and human lens. Exp Eye Res. 2017;161:193-204.
  29. Wang F, Ma J, Han F, Guo X, Meng L, Sun Y, et al. DL-3-n-butylphthalide delays the onset and progression of diabetic cataract by inhibiting oxidative stress in rat diabetic model. Scientific reports. 2016;6:19396.
  30. Chan CY, Guggenheim JA, To CH. Is active glucose transport present in bovine ciliary body epithelium? Am J Physiol Cell Physiol. 2007;292(3):C1087-93.
  31. Zinman B, Wanner C, Lachin JM, Fitchett D, Bluhmki E, Hantel S, et al. Empagliflozin, Cardiovascular Outcomes, and Mortality in Type 2 Diabetes. N Engl J Med. 2015;373(22):2117-28.
  32. Cherney DZ, Perkins BA, Soleymanlou N, Maione M, Lai V, Lee A, et al. Renal hemodynamic effect of sodium-glucose cotransporter 2 inhibition in patients with type 1 diabetes mellitus. Circulation. 2014;129(5):587-97.
  33. Chao EC, Henry RR. SGLT2 inhibition--a novel strategy for diabetes treatment. Nat Rev Drug Discov. 2010;9(7):551-9.
  34. Sanchez-Rangel E, Inzucchi SE. Metformin: clinical use in type 2 diabetes. Diabetologia. 2017;60(9):1586-93.
  35. Walker JM, Eckardt P, Aleman JO, da Rosa JC, Liang Y, Iizumi T, et al. The effects of trans-resveratrol on insulin resistance, inflammation, and microbiota in men with the metabolic syndrome: A pilot randomized, placebo-controlled clinical trial. J Clin Transl Res. 2019;4(2):122-35.
  36. Crandall JP, Oram V, Trandafirescu G, Reid M, Kishore P, Hawkins M, et al. Pilot study of resveratrol in older adults with impaired glucose tolerance. J Gerontol A Biol Sci Med Sci. 2012;67(12):1307-12.
  37. Bhatt JK, Thomas S, Nanjan MJ. Resveratrol supplementation improves glycemic control in type 2 diabetes mellitus. Nutr Res. 2012;32(7):537-41.
  38. Korsholm AS, Kjaer TN, Ornstrup MJ, Pedersen SB. Comprehensive Metabolomic Analysis in Blood, Urine, Fat, and Muscle in Men with Metabolic Syndrome: A Randomized, Placebo-Controlled Clinical Trial on the Effects of Resveratrol after Four Months' Treatment. Int J Mol Sci. 2017;18(3).
  39. Cheng PW, Ho WY, Su YT, Lu PJ, Chen BZ, Cheng WH, et al. Resveratrol decreases fructose-induced oxidative stress, mediated by NADPH oxidase via an AMPK-dependent mechanism. Br J Pharmacol. 2014;171(11):2739-50.
  40. Sadi G, Sahin G, Bostanci A. Modulation of Renal Insulin Signaling Pathway and Antioxidant Enzymes with Streptozotocin-Induced Diabetes: Effects of Resveratrol. Medicina (Kaunas). 2018;55(1).
  41. Sun X, Cao Z, Ma Y, Shao Y, Zhang J, Yuan G, et al. Resveratrol attenuates dapagliflozin-induced renal gluconeogenesis via activating the PI3K/Akt pathway and suppressing the FoxO1 pathway in type 2 diabetes. Food Funct. 2021;12(3):1207-18.
  42. Dludla PV, Silvestri S, Orlando P, Gabuza KB, Mazibuko-Mbeje SE, Nyambuya TM, et al. Exploring the Comparative Efficacy of Metformin and Resveratrol in the Management of Diabetes-associated Complications: A Systematic Review of Preclinical Studies. Nutrients. 2020;12(3).

Reviewer 2 Report

Chen et al investigated the role of SGL2 inhibitors on Fructose-Induced Diabetes Mellitus using samples from human subjects and rat samples.

1.       First, why did the authors select aged patients to test the drugs? How did the authors consider/classify these participants’ cataracts were induced by fructose?  

2.       What were the HbA1c levels of the control subjects?

3.       Using cytosolic Nrf2 alone may not be sufficient to make any conclusions about changes in Nrf2’

4.       Most of the internal control was inconsistent, which may be not used as a control to normalize this dataset. All the immunoblot data may require a repetition with a new set of samples.

The authors may recheck the Immunoblot quantification data also, they are not matching with the raw immunoblot data. 

Author Response

Comments and Suggestions for Authors

Chen et al investigated the role of SGL2 inhibitors on Fructose-Induced Diabetes Mellitus using samples from human subjects and rat samples.

  1. First, why did the authors select aged patients to test the drugs? How did the authors consider/classify these participants’ cataracts were induced by fructose?  

Reply:    

Thank you for your valuable comment.

First of all, the reviewer`s question about why did the authors select aged patients to test the drugs? Diabetic patients aged above 65 may suffer irreversible cataract development; however, good control over metabolism may reverse cataract development in younger diabetic patients (1). Currently, surgery for cataract removal and intraocular lens implants are the main treatments for diabetic cataracts. However, surgery may result in severe postoperative complications, including infection, corneal edema, and ocular hypertension, especially in the elderly and those with hyperglycemic conditions (2). In consequence, additional therapeutic strategies must be developed for the prevention and treatment of diabetic cataracts. Therefore, the studies selected DM (-) patients mean age was 64.0 ± 4.9 years old, and DM (+) patients mean age was 67.0 ± 3.5 years old (P = 0.13).

Second of all, how did the authors consider/classify these participants’ cataracts were induced by fructose?  We did not classify whether these participants` DM were induced by fructose. The Wisconsin Epidemiologic Study of Diabetic Retinopathy reported that 24.9% of type 2 diabetes, 8.3% of type 1 diabetes patients had a history of 10-year-long cumulative incidence for cataract surgery (2). At the end of 2010, the World Health Organization warned that high fructose consumption, mainly in the form of sweetened beverages, is a risk factor for several metabolic diseases (3). Fructose-fed rats are a model of acquired systolic hypertension that displays numerous features of metabolic syndrome in humans. Since all of the data and clinical specimens from Biobank were anonymized before we could access it for analysis, we did not have information regarding fructose consumption for these participants. As a result, these are limitations of this study. Therefore, we set up experiments to confirm our hypothesis on fructose-induced DM.

  1. What were the HbA1c levels of the control subjects?

Reply:    

Thank you for your comment. HbA1c is one of 3 hemoglobin molecules that makes up red blood cells. Glucose attaches slowly to this molecule of hemoglobin over 120 days. Glucose will attach to hemoglobin based on the amount of glucose available. Thus, a glycosylated hemoglobin lab value provides average blood glucose levels of over 3-month period. The HBA1c is an important part of long-term blood glucose monitoring. However, in this study, fructose feeding time period was only 8 weeks. Therefore, the HbA1c levels did not have any difference (as shown in the next page Table 1).

Control (N=5)

Fructose 8 week (N=5)

Mean±SEM

Mean±SEM

HbA1c (%)

4.5 ± 0.04

4.7 ± 0.10

** p<0.001 vs WKY group.

  1. Using cytosolic Nrf2 alone may not be sufficient to make any conclusions about changes in Nrf2’

Reply:    

Thank you for your comment. We agree with the comment about cytosolic Nrf2 alone may not be sufficient to make any conclusions about changes in Nrf2. We modified the sentence to 〝On the contrary, Metformin or Resveratrol inhibited p47-phox, GLUT5, and SGLT2 expressions, but not nuclear factor erythroid 2–related factor 2 (NRF2).〞in the abstract and result 3.4 (Furthermore, Metformin decreased SGLT2 and NADPH oxidase p67-phox protein expressions, and NRF2 was not affected (Fig. 3C-3D) in the revised manuscript. (P.1, Line 32-34; P.7, Line 257-258)

  1. Most of the internal control was inconsistent, which may be not used as a control to normalize this dataset. All the immunoblot data may require a repetition with a new set of samples. The authors may recheck the Immunoblot quantification data also, they are not matching with the raw immunoblot data. 

Reply:

We agree with the reviewer`s point about beta-actin levels. Therefore, we have used different methods (fluorescence and immunoblotting analysis) to estimate the SGLT2, GLUT1/5, RAGE, NOX1/2/4, and p67 levels in the rat lens epithelial sections. However, we have research funding and time limitation for this study. Therefore, in the future, we will repeat experiments with a new set of samples to check all the immunoblot data to improve the quality of our work. Here, we provided all the raw immunoblot data in the following:

References:

  1. Srinivasan S, Raman R, Swaminathan G, Ganesan S, Kulothungan V, Sharma T. Incidence, Progression, and Risk Factors for Cataract in Type 2 Diabetes. Invest Ophthalmol Vis Sci. 2017;58(13):5921-9.
  2. Kumar PA, Reddy PY, Srinivas PN, Reddy GB. Delay of diabetic cataract in rats by the antiglycating potential of cumin through modulation of alpha-crystallin chaperone activity. J Nutr Biochem. 2009;20(7):553-62.
  3. Maarman GJ, Mendham AE, Lamont K, George C. Review of a causal role of fructose-containing sugars in myocardial susceptibility to ischemia/reperfusion injury. Nutr Res. 2017;42:11-9.

Round 2

Reviewer 2 Report

Author's response to reviewer comments in the revision was not satisfactory and the immunoblot data were not consistent within the samples of each group and were not updated in the revised version.